# COVID-19 Vaccine Priority Strategy Using a Heterogenous Transmission Model Based on Maximum Likelihood Estimation in the Republic of Korea

**DOI:** 10.3390/ijerph18126469

**Published:** 2021-06-15

**Authors:** Youngsuk Ko, Jacob Lee, Yeonju Kim, Donghyok Kwon, Eunok Jung

**Affiliations:** 1Department of Mathematics, Konkuk University, Seoul 05029, Korea; kys1992@konkuk.ac.kr; 2Division of Infectious Disease, Department of Internal Medicine, Hallym University College of Medicine, Chuncheon 24252, Korea; litjacob@chol.com; 3Division of Public Health Emergency Response Research, Korea Disease Control and Prevention Agency, Cheongju 28159, Korea; ykim8772@korea.kr (Y.K.); vethyok@korea.kr (D.K.)

**Keywords:** mathematical modeling, COVID-19, vaccine priority, reproductive number, maximum likelihood estimation, healthcare worker

## Abstract

(1) Background: The vaccine supply is likely to be limited in 2021 due to constraints in manufacturing. To maximize the benefit from the rollout phase, an optimal strategy of vaccine allocation is necessary based on each country’s epidemic status. (2) Methods: We first developed a heterogeneous population model considering the transmission matrix using maximum likelihood estimation based on the epidemiological records of individual COVID-19 cases in the Republic of Korea. Using this model, the vaccine priorities for minimizing mortality or incidence were investigated. (3) Results: The simulation results showed that the optimal vaccine allocation strategy to minimize the mortality (or incidence) was to prioritize elderly and healthcare workers (or adults) as long as the reproductive number was below 1.2 (or over 0.9). (4) Conclusion: Our simulation results support the current Korean government vaccination priority strategy, which prioritizes healthcare workers and senior groups to minimize mortality, under the condition that the reproductive number remains below 1.2. This study revealed that, in order to maintain the current vaccine priority policy, it is important to ensure that the reproductive number does not exceed the threshold by concurrently implementing nonpharmaceutical interventions.

## 1. Introduction

The severe acute respiratory syndrome-coronavirus 2 (SARS-CoV-2, COVID-19) pandemic has been spreading worldwide since the end of 2019 from Wuhan, China. As of 6 May 2021, the global cumulative incidence and number of deaths caused by COVID-19 reached more than 155 million and 3 million, respectively [1]. Owing to its characteristics of high transmissibility and severity, COVID-19 has threatened the healthcare systems in many countries, and the immediate nonpharmaceutical interventions that have been proposed to prevent the breakdown of the medical system have posed a serious socio-economic burden [2,3].

These nonpharmaceutical interventions, such as social distancing, wearing masks, washing hands, maintaining personal hygiene, minimizing unnecessary meetings, restricting travel, and remote schooling, are essential for preventing and reducing the transmission of COVID-19. However, they have not been able to completely stop its spread. The transmission of COVID-19 could be ended once the population achieves herd immunity from the pharmacological intervention of vaccination.

Vaccine development has been conducted by several pharmaceutical companies. The mRNA-based vaccine, BNT162b2, acquired emergency-use authorization in December 2020, and the first vaccinations began in England [4]. However, the current vaccine production ability is currently insufficient to halt the global COVID-19 pandemic. Therefore, it is important to minimize the possibility of medical collapse by effectively distributing vaccines. In Korea, vaccination began on 26 February 2021, and the frontline health workers (HCWs) and the elderly in nursing homes and nursing hospitals were prioritized to minimize mortality [5]. The Korean government planned to administer more than 13 million doses, which consist of AZD1222 and BNT162b2, in the second quarter of 2021 [6].

The UK government Vaccine Task Force decided to prioritize vaccination to senior citizens (over 50 years old), to those with comorbidities, and to the HCWs and front-line social workers of the COVID-19 response [7]. Moreover, the reduction of mortality was expected to be the highest when vaccination started at the older ages and expanded to younger ages, especially in developed countries [8].

On 6 May 2021, there were 124,879 patients in Korea, and the number of deaths was 1847 (a mortality rate of 1.48%). The mortality rate for patients above the age of 80 years old was 18.79%, and that of those in their seventies was 5.85%. Patients who died from COVID-19 infection while being admitted to nursing homes or nursing hospitals accounted for approximately 40% of all deaths [9,10].

In the ongoing COVID-19 pandemic, a mathematical model can be used as a scientific policy basis by policy makers. For example, the Center for Disease Control and Prevention in the U.S. adopted mathematical modeling for pandemic planning and resource allocation [11]. In the U.S., Bubar et al. introduced model-informed COVID-19 vaccine prioritization by age and serostatus [12]. They intended that vaccinating adults older than 60 years would minimize mortality.

In Korea before the COVID-19 epidemic, Kim and Jung developed an age-dependent mathematical model of influenza and prioritized the vaccination order to minimize the incidence number. They showed that, although the vaccination policy of the government was effective, prioritizing adults over the elderly might be a better strategy [13]. Following the COVID-19 epidemic, Shim developed an age-dependent model to prioritize COVID-19 vaccination in Korea [14]. Using the POLYMOD contact matrix, Shim found that to minimize mortality, it was best to vaccinate the elderly first.

In this study, we developed a transmission matrix using maximum likelihood estimation (MLE), based on the epidemiological records of individual COVID-19 cases in Korea, instead of using an influenza-based transmission matrix, which was used in other studies. To our best knowledge, this study is the first attempt to develop a transmission matrix using MLE. A heterogenous population model considering HCWs, underage, adult, and elderly groups was developed using this transmission matrix. This matrix was used to analyze the community spread of outbreaks of COVID-19 in nursing homes and nursing hospitals. Finally, from the model simulation results, we provide insight into vaccination prioritization.

## 2. Materials and Methods

### 2.1. Data

The numbers of daily incidents and quarantines were aggregated from the daily press releases of the Korea Disease Control and Prevention Agency (KDCA) from 1 October 2020 to 25 February 2021 [15]. The personal data of COVID-19 patients regarding the age, diagnosis date, and symptom onset date were provided by the Korea Disease Control and Prevention Agency (KDCA). These data also contain the information on whether a patient is an HCW.

### 2.2. Maximum Likelihood Estimation

We used MLE to estimate the transmission rates between heterogenic population groups [16]. The population was divided into four groups considering the heterogeneity of transmission between different age groups and the potential risks of an HCW: HCW, underage (under 19 years), adult (exclude HCWs), and senior (over 65 years) groups. These groups are denoted as 1 (HCW), 2 (underage), 3 (adult), and 4 (senior), respectively, and are generalized using the *X* or *Y* notation. Let $\beta_{XY}$ be the transmission rate from group *Y* to *X*. By assuming that the transmission probability follows a Poisson distribution, with the mean value $\sum \beta_{XY}I_{Y}\left(t\right)/N$, the *i*th individual in group *X* at time *t* has the possibility of staying uninfected until the next time (t+1) as
$$p_{sur,X,i}\left(t\right)=exp\left(-\frac{\sum \beta_{XY}I_{Y}\left(t\right)}{N}\right),$$
where *N* is the total number of the population and $I_{Y}\left(t\right)$ denotes the number of infectious hosts in group *Y* at time *t*. On the contrary, the possibility that the *i*’th individual in group *X* is exposed at time *t* and becomes infected at t+1 is
$$p_{inf,X,i}\left(t\right)=1-exp\left(-\frac{\sum \beta_{XY}I_{Y}\left(t\right)}{N}\right).$$

$X_{S}$ and $X_{I}$ denote the uninfected and infected subgroups of *X*, respectively. Therefore, the likelihood of every group’s (*L*), which reflects the individual events (being infected or straying uninfected), is formulated as follows:$$L=\prod_{X}\left\{\prod_{i\in X_{I}}\left[\left(\prod_{j=0}^{t_{inf,X_{I},i}-2}p_{sur,X_{I},i}\left(j\right)\right)p_{inf,X_{I},i}\left(t_{inf,X_{I},i}-1\right)\right]\prod_{k\in X_{S}}\left(\prod_{j=0}^{t_{f}-1}p_{sur,X_{S},k}\left(j\right)\right)\right\}.$$ Time when the *i*th individual in group $X_{I}$ is infected is $t_{inf,X_{I},i}$, and $t_{f}$ represents the final time. The likelihood is a function of the transmission rates, which can be numerically estimated by finding a set that maximizes the likelihood.

### 2.3. Heterogeneous Population Model of COVID-19 Epidemic in Republic of Korea

In this section, we present the development of a heterogeneous population model of COVID-19 using the estimated transmission rates. A flowchart of the model is displayed in Figure 1. *S* is a susceptible group, and a host in this group can be infected by infectious hosts. A susceptible host who is infected becomes the host of *E*, which is an exposed group. The hosts in *E* still cannot infect the susceptible hosts.

Subsequently, a host in *E* becomes a host in *I*, which is an infectious group, and can infect the susceptible hosts. *Q* is an isolated group, and it is assumed that the isolated hosts can no longer infect the others. *R* represents a recovered group, and the hosts in this group are immune to the disease. We introduce the heterogeneous population model of COVID-19 based on the estimated transmission risk rates using MLE, as discussed in the previous subsection, as follows:$$\begin{array}{cc} \mathrm{For}\;X,\;Y & =1,\;2,\;3,\;4,\\ \frac{dS_{X}}{dt} & =-\beta_{0}\left(t\right)\sum_{Y}\beta_{XY}S_{X}\frac{I_{Y}}{N},\\ \frac{dE_{X}}{dt} & =\beta_{0}\left(t\right)\sum_{Y}\beta_{XY}S_{X}\frac{I_{Y}}{N}-\kappa E_{X},\\ \frac{dI_{X}}{dt} & =\kappa E_{X}-\alpha I_{X},\\ \frac{dQ_{X}}{dt} & =\alpha I_{X}-\gamma Q_{X},\\ \frac{dR_{X}}{dt} & =\left(1-f_{X}\right)\gamma Q_{X},\\ N_{X} & =S_{X}+E_{X}+I_{X}+R_{X},N=\sum N_{X}. \end{array}$$

A time-dependent parameter, $\beta_{0}\left(t\right)$, is considered to reflect the time varying government interventions and behavior changes of the population. The parameters κ and α are the mean values of the progression and quarantine rates, respectively. γ is the mean recovery rate, and $f_{X}$ is the mean fatal rate of group *X*. The progression, quarantine, and recovery rates were acquired from the epidemiological reports and the KDCA press releases [17,18,19].

The transmission rate, $\beta_{XY}$, was estimated using MLE, as discussed in the previous subsection. The estimated transmission rates were the averaged values for a long simulation period from 1 October 2020 to 25 February 2021. The Korean government’s intervention policies changed many times after the outbreaks of COVID-19. As the contact patterns also differed based on the government intervention policy, the transmission rate constant, $\beta_{0}\left(t\right)$, was estimated by dividing the phase to reflect this factor using the least squares fitting toolbox (lsqcurvefit) of MATLAB, while the ordinary equation system was solved using the solver ode45 of MATLAB.

In this procedure, we excluded border screening cases from the data, but included the cases from abroad that were diagnosed in the local community. The model simulation started on 1 October 2020 before the third wave began, and the final time was set as 25 February 2021, because the vaccination began on 26 February 2021. The initial values of $\left(I_{1},I_{2},I_{3},I_{4}\right)$ and $\left(E_{1},E_{2},E_{3},E_{4}\right)$ were estimated by the following procedure: on 1 October 2021, anyone infected before 1 October 2021, but whose serial interval was not yet started was set as an exposed host.

Concurrently, if his or her serial interval had already started but not yet ended, the person was set as an infectious host. Using the personal data of the COVID-19 patients, the estimated initial values of $\left(E_{1},E_{2},E_{3},E_{4}\right)$ and $\left(I_{1},I_{2},I_{3},I_{4}\right)$ were (5,16,85,23) and (12,18,249,81), respectively. We used the population number of each group excluding the number of infected as the initial values of $\left(S_{1},S_{2},S_{3},S_{4}\right)$. The HCW population was estimated using the number of HCWs per 100,000 statistics, and this number was excluded from the adult population [20,21].

The reproductive number is the expected number of secondary cases by the first infectious host in the susceptible group during the mean infectious period. Using the next-generation method, a 4 × 4 next generation matrix, *G*, was formulated [22]. The elements of the next-generation matrix are denoted by i,j=1,2,3,4, as follows,
$$G\left[i,j\right]=\beta_{0}\left(t\right)\frac{\beta_{ij}}{\alpha }\frac{S_{i}}{N}.$$ Elements of the next-generation matrix are the reproductive numbers between the coupled groups. A reproductive number was calculated by finding the maximum eigenvalue of the matrix, *G*. More details are in [22].

### 2.4. Vaccine Priority Strategy

We present the construction of a model of COVID-19 in Korea considering the vaccine effectiveness. We adapted the vaccination model developed in a previous study and added three more compartments related to vaccination to our developed model [13]. When a host is effectively vaccinated, he or she moves to compartment $V_{X}$—a group of hosts who have been effectively vaccinated. However, $V_{X}$ is still non-immune to the disease, and therefore the hosts in this group can be infected before they become immune.

Compartment $U_{X}$ represents a group of hosts who are vaccinated ineffectively, and $P_{X}$ represents a protected group that is immune to the disease. Figure 2 shows a flowchart of the vaccine-applied model. We set that 86.6% of vaccinated individuals acquire immunity after 14 days [6]. The following 32 differential equations represent the heterogeneous population model system considering vaccination: $$\begin{array}{cc} \mathrm{For}\;X,\;Y & =1,\;2,\;3,\;4,\\ \frac{dS_{X}}{dt} & =-\beta_{0}\sum_{Y}\beta_{XY}S_{X}\frac{I_{Y}}{N}-Vacc,\\ \frac{dE_{X}}{dt} & =\beta_{0}\sum_{Y}\beta_{XY}\left(S_{X}+U_{X}+V_{X}\right)\frac{I_{Y}}{N}-\kappa E_{X},\\ \frac{dI_{X}}{dt} & =\kappa E_{X}-\alpha I_{X},\\ \frac{dQ_{X}}{dt} & =\alpha I_{X}-\gamma Q_{X},\\ \frac{dR_{X}}{dt} & =\left(1-f_{X}\right)\gamma Q_{X},\\ \frac{dU_{X}}{dt} & =\left(1-e\right)*Vacc-\beta_{0}\sum_{Y}\beta_{XY}U_{X}\frac{I_{Y}}{N},\\ \frac{dV_{X}}{dt} & =e*Vacc-\beta_{0}\sum_{Y}\beta_{XY}V_{X}\frac{I_{Y}}{N}-\omega V_{X},\\ \frac{dP_{X}}{dt} & =\omega V_{X},\\ N_{X} & =S_{X}+E_{X}+I_{X}+R_{X}+U_{X}+V_{X}+P_{X},N=\sum N_{X}. \end{array}$$

To prioritize the vaccination order, we consider various vaccination scenarios. Note that we use *M* to count millions. First, we set up three cases of total vaccine dose per group. In all scenarios, the total doses for HCWs is fixed as 0.5M because the total number of HCWs in Korea is less than 0.6M. The dosages for the other groups are set as 1M, 1.5M, or 2M. The daily number of doses is set as 0.1M or 0.2M. The reproductive numbers are set from 0.1 to 2.0 in 0.1 intervals (20 cases). Therefore, we consider 120 scenarios in total. The model simulation is extended by 100 days in each scenario, and vaccination starts from the beginning of the extended days. Our objective was to determine the best vaccination priority to minimize the cumulative mortality or the cumulative incidence.

## 3. Results

### 3.1. Maximum Likelihood Estimation

The transmission rates estimated by MLE are listed in Table 1 and visualized in Figure 3. The transmission rate was the highest among HCWs (8.33), followed by the among seniors (0.64). The transmission rates among underage hosts and for adult-to-HCW were the third (0.52) and fourth highest (0.48), respectively.

We adjusted the transmission rates considering the population sizes of the groups to determine the transmission risk. The adjusted transmission risks for disease transmission and being infected are displayed as bar graphs in Figure 4. Without considering the to-same or to-other-groups, the adult hosts had the highest risk of disease transmission (0.23). The HCW group had the second highest risk of disease transmission (0.14) but the highest risk of disease transmission to-other-groups (0.05). The hosts in the underage and senior groups have relatively lower risks of disease transmission compared with the hosts in the adult and HCW groups.

The HCWs had the highest risk of being infected (0.09 by themselves and 0.33 by the other groups). The adult and senior groups had opposite types of risks of being infected. However, their summed values were similar. For the adult group, most of (more than 99.99%) the risk of being infected was caused by its own hosts. However, for the senior group, only 55.63% of the risk of being infected was caused by its own hosts.

### 3.2. Parameterization of the Mathematical Model

The model system was used to estimate the time-dependent $\beta_{0}\left(t\right)$. Figure 5 displays the best fitted model curves, data for the daily incidence (B), cumulative incidence (C), and reproductive numbers (A) according to the government intervention policy. Here, we only plot the total cumulative incidence and not the group-dependent incidence, and the simulation duration was from 1 October 2020 to 25 February 2021.

Panel (A) shows the reproductive numbers according to the government’s intervention policy. The red dotted line represents the threshold reproduction number value (1). The maximum reproductive number was 1.36, from 12 October to 4 November 2020 when the government’s social distancing level was at the lowest level. The minimum value was 0.69, from 23 December 2020 to 18 January 2021 when the ban on private gatherings of five or more people was pronounced. The red squares denote the real data, and the dark curves are the fitted model results. Note that the cumulative number contains cases before 1 October 2020. The fitted values of $\beta_{0}\left(t\right)$ are listed in Table 2, and the other model parameters are listed in Table 3.

Figure 6 shows a pie chart of the cumulative incidence obtained from the model simulation. The number of COVID-19 incidents from 1 October 2020 to 25 February based on the model simulation was 62,530, and the numbers of HCW, underage, adult, and senior cases were 2479, 12,149, 41,219, and 6683, respectively. The proportion of adult cases in the total cases was the highest (65.9%). However, when the incidence number was adjusted by the population size of each group (panel B), the HCW group had the highest number of incidence of 426.5 per 100 K, which was more than three-fold that of the adult group (118.1 per 100 K).

### 3.3. Vaccine Priority

There are 4!=24 possible permutations of the vaccine priority order for each scenario based on the heterogeneous population model considering vaccination. The simulation results of the observed 120 scenarios show that there are six unique vaccination orders that minimize the cumulative incidence or mortality: 1-4-3-2, 1-3-4-2, 3-1-4-2, 4-1-3-2, 4-3-1-2, and 3-4-1-2. Figure 7 shows the color-marked vaccination orders of the 120 scenarios, and panels (A) and (B) display the vaccination orders minimizing the cumulative incidence and mortality, respectively.

The marked colors represent the unique vaccination orders. Note that the simulation period is 100 days because we are considering a short-term vaccine priority strategy. For the objective of minimizing the mortality (panel B), the vaccination order 4-1-3-2 is the best strategy for relatively low reproductive numbers of the same or less than 1.2. However, if the reproductive number is higher than 1.2, the vaccination order of 3-4-1-2 is the best to minimize the number of deaths.

This result suggests that, to reduce the mortality, the best strategy is to vaccinate the elderly and the HCWs first. To minimize the number of cumulative incidence (panel A), for all scenarios with a reproductive number greater than or equal to 0.9, the order of 3-1-4-2 is the best vaccination strategy. If the reproductive number is very low (0.1 to 0.2), the vaccination order of 1-4-3-2 is the best strategy, and if it is 0.3 to 0.8, the vaccination order of 1-3-4-2 is best. These results show that first vaccinating the adult group, who has a high risk of social contact, is an optimal strategy to minimize the number of cumulative incidents.

Table 4 summarizes the simulation results for different vaccination orders in the same scenario: the vaccine supply for all groups is (0.5, 1.5, 1.5, and 1.5)*M* with a daily dose number of 0.1*M* and reproductive number of 1.2. As Figure 7 shows, the 3-1-4-2 vaccination order minimizes the number of cumulative incidence by reducing it by 27% compared to that without vaccination. When the 4-1-3-2 vaccination order is applied, the reduction in the cumulative incidence is relatively small (reduction: 22.24%) compared to that when 3-1-4-2 is applied; however, the mortality is minimized by 1901 (reduction: 25.96%).

## 4. Discussion

We investigated the best vaccine prioritization strategy for COVID-19 to minimize the incidents or deaths using a heterogenous population model considering vaccination in Korea. In this study, the transmission matrix was constituted using real data of the incident cases of COVID-19, instead of by a survey as previous studies have done. The novelty of this study is that the transmission matrix was constructed using the MLE method based on the data of symptom onset and the confirmed dates of patients. The transmission rates of the four heterogenous groups were estimated by multiplying the transmission matrix by a constant according to the changes in the Korean government policy. The mathematical model simulations using the transmission matrix and the phase-dependent adjusting constant realistically demonstrated the transmission dynamics of the COVID-19 epidemic in Korea.

The estimated transmission matrix showed that HCWs had a relatively higher risk of being infected compared with the other groups and that adults play a major role in community spread. The mass outbreak events at the elderly care facilities were also reflected in the matrix by the higher transmission rates among the senior group hosts and HCW-to-senior rates in Korea [23]. The above results emphasize the need to provide protective equipment to HCWs and the importance of blocking external infections to prevent mass infections in elderly care facilities [24].

In contrast, the underage group had the lowest infection risk among the four groups. This is related to the significant decrease in mass gathering events due to school opening delays, the starting of online classes, and attendance restrictions [25]. Most of the incidence occurs in the adult group; however, when it is adjusted by the population size of each group, the HCWs and the seniors had relatively higher incidence numbers. The incidence per 100,000 of HCWs from the model simulation was 426.5 in Korea, which is more than three-fold the adult population group but relatively smaller than those in the European countries or the U.S. For example, in the U.S., the incidence per 100,000 of HCWs was approximately four-fold that of a normal community [26].

With a limited supply of vaccines in all countries, who should be vaccinated first for the best overall vaccination effect is an important issue. In this study, using a heterogenous population model considering vaccination, we found that the vaccine priority changed depending on the objective (minimizing the cumulative incidence or the cumulative mortality) and the effective reproductive number. The priority also changed according to the total and daily doses, whereas it was comparatively less sensitive to the reproductive number. Prioritizing vaccination of the source of infection, which was the adult group in this study, was the best strategy to minimize the incidence.

This is similar to the results of previous studies on COVID-19 and the 2009 A/H1N1 influenza pandemic [13,14]. Due to the high risk of COVID-19 to the elderly group, vaccinating a vulnerable host (the elderly group) and a host with high connectivity to a vulnerable host (the HCW group) was the best strategy to minimize the mortality when the reproductive number is below a certain threshold.

In Korea, to minimize mortality, vaccination was planned with priority to HCWs and senior citizens [5]. This was because they have a relatively higher risk of infection owing to the characteristics of the groups. As many of elderly people are in facilities, such as nursing homes, the spread of infection in these facilities has serious consequences. Our model simulation also reflected these points and yielded results in agreement with the Korean policy.

The effective reproductive number on 26 February 2021, which was the first day of vaccination, was approximately 1.15. However, for a reproductive number exceeding 1.2, our simulation showed that vaccinating the adult group first, which was the main source of infection, could reduce the mortality. To maintain the current Korean government’s vaccination policy, paradoxically, it is necessary to continue strong nonpharmaceutical interventions, such as enhanced social distancing during vaccination, to ensure that the reproductive number remains below the threshold.

A transmission matrix can be generated from surveys. The POLYMOD contact pattern in the study was based on contact surveys conducted in eight European countries [27]; however, there were no survey data for most of countries, including Korea. Developing the transmission matrix using the MLE based on the personal data of patients as done in this study will be a useful and innovative tool for investigating the transmission dynamics of emerging or infectious diseases.

Vaccine hesitancy, which refers to a delay in acceptance or the refusal of vaccines despite the availability of vaccine services, is one of the important factors during vaccination [28]. However, in Korea, as the vaccine acceptance rate of healthcare workers and the elderly is more than 90%, we did not consider vaccine hesitancy in this study [29]. There are some limitations to this study. First, the transmission matrix was constructed based on the dates of symptom onset and diagnosis, assuming that social contact patterns were implicitly included in the model parameters and the transmission rate according to the Korean government’s policy. Second, our study did not consider a second dose, which enhances the vaccine effectiveness. A model considering detailed age groups and vaccines will be studied in a future study.

## 5. Conclusions

In this study, we divided the population of Korea into four heterogenous groups and estimated the transmission rates between the groups using the MLE method. A mathematical model of the COVID-19 epidemic considering vaccination in Korea was constructed using these estimated transmission rates. We determined the best vaccination priority order by minimizing either the incidence or mortality for the early vaccination period. Intuitively, prioritizing the source of infection (adult) was hypothesized to be the best strategy to minimize the incidence number, and prioritizing hosts who are vulnerable to disease (the elderly) or close to vulnerable hosts (HCWs) was hypothesized to be the best strategy to minimize mortality.

Our model simulation showed similar results as prioritizing the adult group was best to minimize the incidence number when the reproductive number was over 0.9. On the other hand, HCWs and elderly groups should be prioritized to be vaccinated first if the initial objective of vaccination is to minimize the mortality when the reproductive number is below 1.2. The results of our study support the current Korean government policy to minimize the mortality and prioritize the vaccination of HCWs and the elderly. Note that, as of 26, February 2021, the estimated reproductive number in Korea was below the threshold [30].

However, our simulations showed that if the COVID-19 situation worsens, vaccinating the adult group might be a better strategy than the present one, although the object is still minimizing mortality, not incidence. However, this opposes the demand for strong nonpharmaceutical interventions with an immediate response when the reproductive number reaches the threshold or more. Therefore, it would be best to prioritize the vaccination of HCWs and the elderly while continuing to socially suppress COVID-19 to a stable level.

## Figures and Tables

**Figure 1 ijerph-18-06469-f001:**
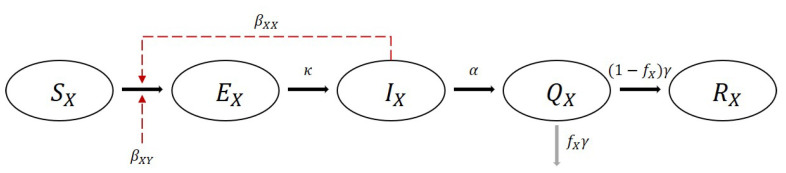
Flowchart of the mathematical model of the COVID-19 epidemic in Korea considering group heterogeneity.

**Figure 2 ijerph-18-06469-f002:**
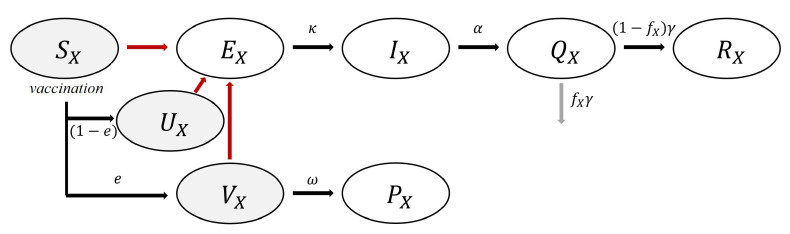
Flow chart of vaccination-applied mathematical model.

**Figure 3 ijerph-18-06469-f003:**
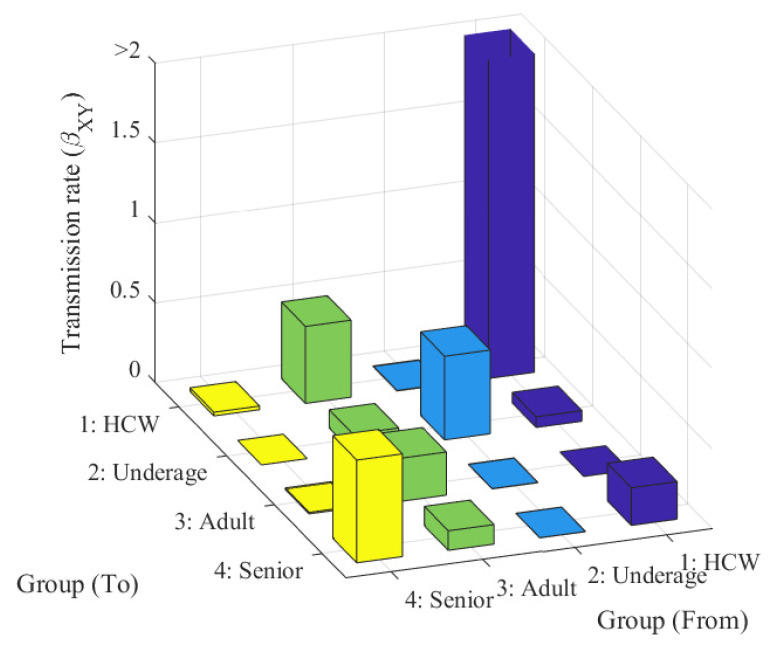
Estimated transmission rates. Purple, blue, green, and yellow represent the HCW, underage, adult, and senior groups, respectively.

**Figure 4 ijerph-18-06469-f004:**
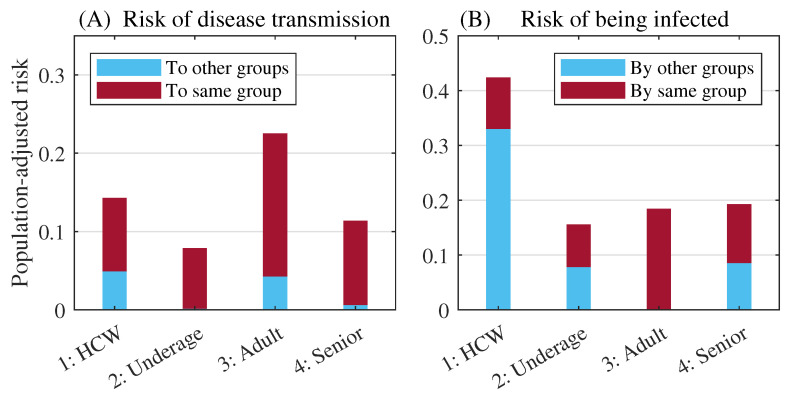
The population-adjusted risk of disease transmission (**A**), and risk of being infected (**B**). Cyan and red boxes denote extrinsic (from or to different groups) and intrinsic (among same group hosts) factors, respectively.

**Figure 5 ijerph-18-06469-f005:**
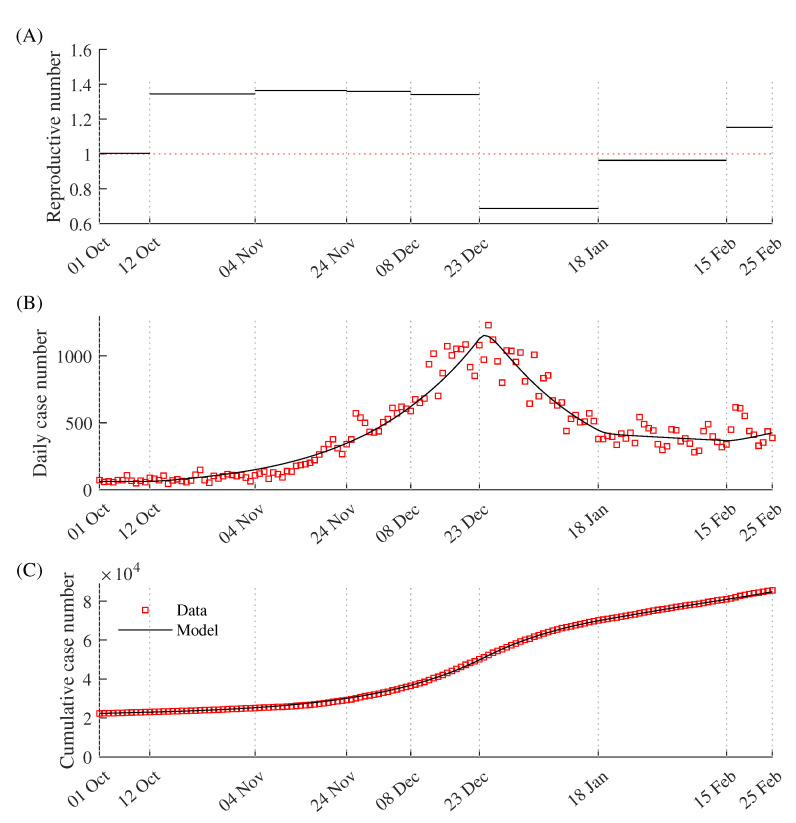
The parameter estimation results. Red squares and dark curves denote the real data and model simulation results using data-fitted parameters, respectively. (**A**) reproductive number, (**B**) daily incidence, and (**C**) cumulative incidence.

**Figure 6 ijerph-18-06469-f006:**
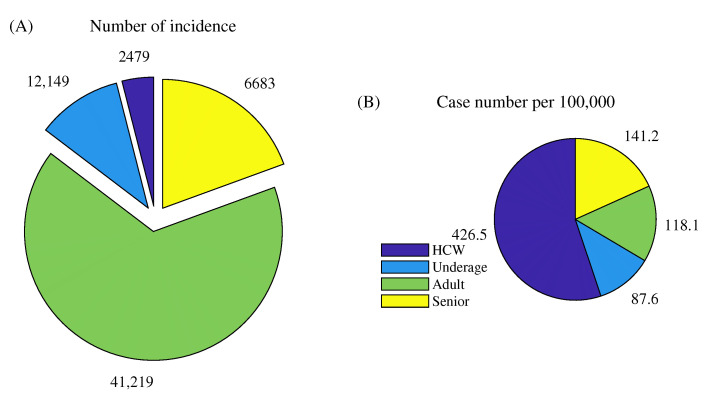
Confirmed case number of groups (**A**), and population-adjusted case number per 100,000 (**B**). Purple, blue, green, and yellow denote the HCW, underage, adult, and senior groups, respectively.

**Figure 7 ijerph-18-06469-f007:**
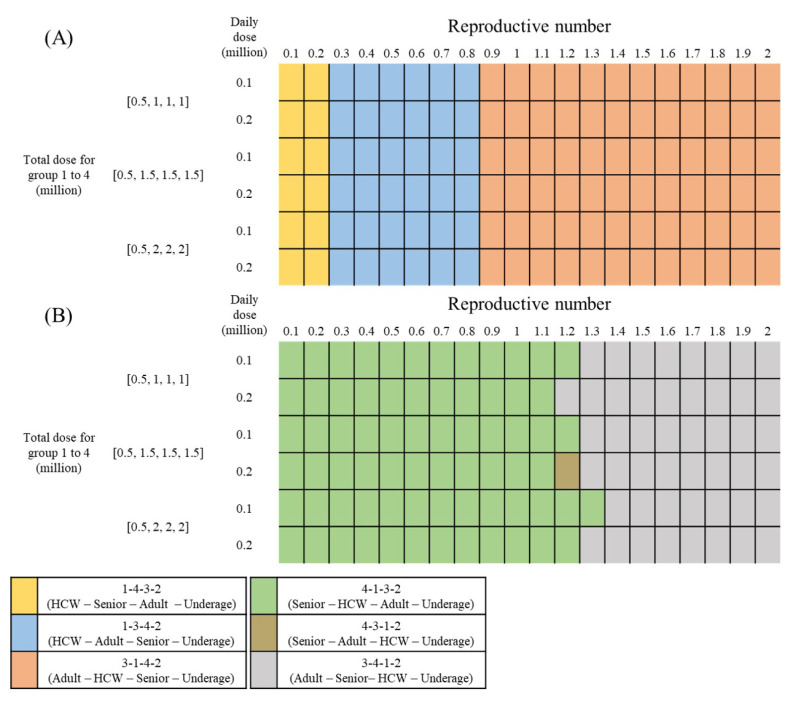
Best vaccination order for 120 scenarios. (**A**) minimizing incidence, and (**B**) minimizing mortality.

**Table 1 ijerph-18-06469-t001:** The estimated transmission rates ($\beta_{XY}$).

			From		
		1: HCW	2: Underage	3: Adult	4: Senior
	1: HCW	8.3315	0.0029	0.4831	0.0230
To	2: Underage	0.0667	0.5246	0.1146	0.0001
	3: Adult	0.0001	0.0019	0.2706	0.0091
	4: Senior	0.2358	0.0010	0.1226	0.6444

**Table 2 ijerph-18-06469-t002:** The data-fitted phase-dependently adjusted constant ($\beta_{0}\left(t\right)$).

Phase	Period	Value
1	1 August 2020–11 August 2020	0.9058
2	12 August 2020–3 November 2020	1.2134
3	4 November 2020–23 November 2020	1.2317
4	24 November 2020–7 December 2020	1.2273
5	8 December 2020–22 December 2020	1.2105
6	23 December 2020–18 January 2021	0.6205
7	18 January 2021–15 February 2021	0.8696
8	15 February 2021–25 February 2021	1.0404

**Table 3 ijerph-18-06469-t003:** Model parameters.

Symbol	Description	Value	Reference
1/κ	average duration from exposed to infectious	2.1 days	[17,18,19]
1/α	average duration from infectious to quarantine	6 days	[17,18,19]
1/γ	average duration from quarantine to recover or death	25 days	[15]
$f_{1}$, $f_{3}$	fatal rate of adult (include HCW)	0.0031	[15]
$f_{2}$	fatal rate of underage	0	[15]
$f_{4}$	fatal rate of senior	0.1072	[15]

**Table 4 ijerph-18-06469-t004:** Simulation results depending on the different vaccination orders in the same scenario (vaccine supply for each group: (0.5, 1.5, 1.5, and 1.5)*M*, daily dose number: 0.1M, and reproductive number: 1.2).

Vaccination Order	Additional Case (Reduction)	Additional Fatality (Reduction)
without vaccination	170,389	2568
1-4-3-2	132,097 (22.47%)	1907 (25.75%)
1-3-4-2	126,104 (25.99%)	1934 (24.68%)
3-1-4-2	124,389 (27.00%)	1934 (24.68%)
4-1-3-2	132,489 (22.24%)	1901 (25.96%)
4-3-1-2	131,129 (23.04%)	1906 (25.76%)
3-4-1-2	124,920 (26.69%)	1927 (24.95%)

## Data Availability

The numbers of daily incidence and quarantine were aggregated from the daily press releases of the Korea Disease Control and Prevention Agency (KDCA) from 1 October 2020 to 25 February 2021.

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
