# Peer review of "COVID-19 Vaccine Priority Strategy Using a Heterogenous Transmission Model Based on Maximum Likelihood Estimation in the Republic of Korea"

_ijerph, 2021, doi:10.3390/ijerph18126469_

Round 1
Reviewer 1 Report
The results of this study provide important evidence in support of the Republic of Korea´s vaccination strategy, as well as the vaccination strategy of the majority of countries, if not all, across the world - vaccinate HCWs and the elderly first. The mathematical model used by the authors is adequately described and the results of the simulation are robust. Further, the overall results and conclusions of the study are highly relevant for policy decisions related to vaccination against COVID-19, especially in light of the limited supply of vaccines globally.
However, given that the vaccination strategy discussed in the paper is being implemented widely- which was expected prior to the availability of the vaccines based on the risk of exposure of HCWs and vulnerability and death rates of the elderly compared to other age groups-, it is fair to ask why the authors did not explore other questions that countries may be struggling with regarding whom to vaccinate next. As more vaccines become available, age of elegibility of vaccination is expanded (chilren older than 12), and more data may become available with regards to levels of protection including by gender, for instance, it would have been helpful to run simulations regarding such variables. The authors make a brief reference to this but do not elaborate about it. The paper can be strengthened by providing some insights with regards to these issues.
Another important dimension the authors could consider discussing, even if briefly, is what types of variables may be missing. One such variable could be vaccine hesitancy. What are vaccine hesitancy levels like in Korea in both populations included in the study? Even if vaccine hesitancy figures in Korea are low, it would still be worth considering vaccine hesitancy given how substantial it remais in many countries around the world, with great implications for herd immunity and population wide protection. While the study´s focus is Korea, such consideration would be of great relevance for researchers interested in running similar models and analyses in other countries.
Overall, the paper makes an important contribution to the literature and to policy decisions in Korea concerning vaccine strategy against COVID-19. But, as pointed out above, the authors could enhance the relevance of the paper in Korea and beyond by addressing these recommendations.
Reviewer 2 Report
The paper is well written.
Please answer a few questions you can find in the attached commented paper.
